# The Impact of the COVID-19 Pandemic on Physicians’ Working Hours and Earnings in São Paulo and Maranhão States, Brazil

**DOI:** 10.3390/ijerph191610085

**Published:** 2022-08-15

**Authors:** Bruno Luciano Carneiro Alves de Oliveira, Lucas Salvador Andrietta, Regimarina Soares Reis, Ruth Helena de Souza Britto Ferreira de Carvalho, Maria Teresa Seabra Soares de Britto e Alves, Mário César Scheffer, Giuliano Russo

**Affiliations:** 1Program in Public Health, Federal University of Maranhão, São Luís 65080-805, Brazil; 2Department of Preventive Medicine, University of São Paulo, Piracicaba 13416-000, Brazil; 3Joaquim Venâncio Polytechnic School of Health, Oswaldo Cruz Foundation, Rio de Janeiro 21040-900, Brazil; 4Wolfson Institute of Population Health, Queen Mary University of London, London E1 4NS, UK

**Keywords:** COVID-19, physician health, health workers in Brazil, health system, health labor market, health workforce in LMICs

## Abstract

Evidence exists on the health impacts of the current COVID-19 pandemic on health workers, but less is known about its impact on their work dynamics and livelihoods. This matters, as health workers—and physicians in particular—are a scarce and expensive resource in low- and middle-income countries (LMICs). Our cross-sectional survey set out to explore changes in working hours and earnings during the second year of the pandemic in a representative sample of 1183 physicians in Brazil’s São Paulo (SP) and Maranhão (MA) states. Descriptive analysis and inferential statistics were employed to explore differences in working hours and earnings among public and private sector physicians across the two locations. The workloads and earnings of doctors working exclusively in the public sector increased the most in the second year of the epidemic, particularly in MA. Conversely, the largest proportion of private-only doctors in our sample saw a decrease in their working hours (48.4%, 95% CI 41.8–55.0), whereas the largest proportion of public-only doctors in MA saw an increase in their working hours (44.4%, 95% CI 38.0–50.8). Although earnings remained broadly stable in the public sector, a third of public sector-only physicians in MA saw an increase in their earnings (95% CI 24.4–36.2). More than half of private-only doctors across both states saw a decrease in their earnings (52.2%, 95% CI 45.6–58.8). The largest proportion of dual practitioners (the majority in Brazil and in our sample) maintained their pre-pandemic levels of income (38.8%, 95% CI 35.3–42.3). As public-sector doctors have been key in the fight against the pandemic, it is critical to invest in these cadres in order to develop epidemic preparedness in LMICs, and to find new ways to harness for-profit actors to deliver social benefits.

## 1. Introduction

The recent COVID-19 pandemic has had a substantial impact on the health of populations worldwide. In excess of 515 million cases have been recorded and 6.3 million deaths have been reported at the time of writing [1]. Health workers have been hit particularly hard because of their frontline role in the clinical fight against the pandemic. The World Health Organization estimated that between 80,000 and 180,000 may have died globally because of the virus [2]. Exhaustion, mental health issues and burn-out are some of the associated conditions that have taken their toll on health workforces in high-income [3], middle-income [4] and low-income [5,6] countries.

Less is known about the impact that the crisis has had on health professions and on health workers’ livelihoods. Although epidemics, and the economic recessions that typically follow them, inevitably take their toll on health systems [7,8], recent evidence also suggests that crises also introduce changes in the way health professionals adapt and carry out their healthcare duties [9], and that employment and earning opportunities may not necessarily deteriorate [10]. A recent review of the evidence [11] appears to show that the impact of crises on health workforces is complex and nuanced, as not only do health systems adapt to cope with the changing circumstances, but the demand for health services will also always be sustained during a crisis, guaranteeing employment opportunities for professionals in this area.

This matters, as health workers—and physicians in particular—are a scarce and expensive resource for health systems in low- and middle-income countries (LMICs) [6]. This is especially the case in Brazil, where the health has historically been subsidized, and appears to have shifted some of its elements to adapt to the challenges brought about by the most recent economic crisis [12], which has had varying consequences for the country’s very unequal states and economic conditions [13]. Brazil has a dynamic and complex healthcare system, consisting of a variety of public and private organizations established in different historical moments, which presents problems in terms of the supply and distribution of healthcare services and medical professionals [14,15]. The country’s healthcare system includes the publicly funded Unified Health System (SUS), which is comprehensive and free at the point of delivery, and private healthcare providers, at times funded through public funds and providing services to the SUS. Established in 1990, the SUS is credited to having improved the health of Brazilians by implementing a national primary care program and making a wide range of services available and affordable to the country’s large and diverse population. Nonetheless, the SUS is these days marred by inefficiencies, long waiting lists, inequalities across its diverse states and insufficient funding [16].

Private funds account for approximately 60% of total health spending. Private providers are responsible for several sectors of healthcare provisions, from medical practices to hospitals and laboratories, providing services to both healthcare plan holders and SUS. Access to such private services is mostly granted through health insurance plans, often connected with employment; 24.2% of Brazil’s population owned a private health insurance plan in 2020.

Social Health Organizations are private entities providing health services within SUS. In the last 20 years, low-cost walk-in clinics (People’s Clinics) have started to provide out-of-pocket private services, mostly of an outpatient nature. These People’s Clinics represent an alternative for those who cannot afford a private health plan and wish to see a specialist, but want to avoid SUS’s waiting lists, but they have attracted criticism for adopting a mercantilist approach to provision of care, and for offering poor quality services disconnected from any healthcare reference system.

The COVID-19 pandemic hit Brazil particularly hard: almost 34 million cases have been reported at the time of writing, and around 680,000 deaths. The World Health Organization estimated that 13,525 health workers died in the country because of the pandemic, the world’s second largest loss [2]. Before the pandemic, more than half of Brazilian physicians engaged in dual practice—concurring clinical employment in public and private sector institutions—whereas only one-fifth were exclusively dedicated to public services [17].

This study describes and analyzes the changes experienced by physicians in Brazil during the second year of the COVID-19 crisis, comparing their sectors of activity—public, private and dual practice—in two states with different socioeconomic realities and different levels of access to the health system: Maranhão (MA) and São Paulo (SP). This comparison is relevant because SP is the most populous state and has the second highest gross domestic product (GDP) per capita in the country. The physician labor market is also markedly different in the two states, as SP has 3.2 physicians per 1000 inhabitants [15], and accounts for about 60% of the country’s private health insurance market, covering about 38% of the population (over 50% in the capital). On the other hand, MA has the lowest GDP per capita in the country and has 1.08 doctors per 1000 inhabitants. In MA, 7% of the population has access to private health plans, representing less than 1% of the national market [13]. SP has a balanced relationship between the availability of doctors in the capital and in the cities of the interior, with an inequality indicator equal to 2.43 (the higher this indicator, the greater the inequality). In MA this indicator is 12.99, pointing to a high concentration of physicians in the capital, São Luís [15]. A recent analysis of physician salary data [18] showed, however, that a larger proportion of MA doctors belong to the profession’s highest monthly salary brackets (above R$22,470, or USD 4177) than SP doctors (40.3% and 22.5%, respectively).

Existing evidence highlights the great variability in physicians’ earnings in Brazil across genders, states and specialties [18]. Recent studies have also shown the different ways in which the COVID-19 pandemic has affected the physical and mental health of health professionals in the country, and restructured their work routines, workloads and responsibilities, particularly for frontline physicians and nurses [19]. It has been suggested that physicians and the medical labor market may remain relatively immune from economic crises [20]. It is therefore possible that the impacts of the COVID-19 pandemic on work dynamics, income and working hours may have been felt rather unevenly among Brazilian physicians.

Based on a cross-sectional survey conducted among physicians in the two states, we analyzed the self-reported changes experienced in terms of working hours and earnings in the second year of the pandemic. Ultimately, the paper’s aim is to make an empirical and theoretical contribution to the growing literature on the impacts of crises on human resources for health, and their reactions to crises, particularly in LMICs.

## 2. Materials and Methods

This is a cross-sectional study based on data from a telephone survey conducted with a representative sample of physicians from two states in Brazil (MA and SP). The survey was conducted between 16 February and 15 June 2021, as part of a broad research project on the impacts of COVID-19 and the economic recession on the healthcare system and the medical workforce in one underdeveloped and one industrialized state in Brazil [21]. The nominal listing of physicians registered with Brazil’s Federal Council of Medicine (FCM) in the two states was used to draw the sample. The entire survey was conducted by a specialized company (*Datafolha* Research Institute), under the direct technical supervision of the researcher collaborators.

### 2.1. Data Collection and Sampling Strategy

The study’s overall sample was composed of 1183 physicians, 632 from SP and 551 from MA. The sample was calculated based on the active physicians registered with the FCM in the two states (N = 144,852–152,511 in SP and 7659 in MA), and their key demographic characteristics. The reason why a smaller proportion of the total of physicians in São Paulo was selected in our sample is because the two states are very different—in SP there are 30% of all the physicians in Brazil, and in MA a little more than 1%. Therefore, a proportional sample for each state could not be used, as the MA one would be too small to allow for a sufficient N for some of the strata and doctor characteristics of interest. Two independent samples were instead calculated for the two states, each reflecting the characteristics of interest for our stratification. Comparisons between the physicians in the two states were then adjusted by the samples’ weights, or by using a mixed model, to account for the different variance in the two sets. For the sample calculation, we considered 95% confidence intervals (95% CIs), a margin of error of 5% and statistical power of 80.0% (see the two simple random sampling with replacement equations in the box 1 below).

Substitution was carried out in cases of unsuccessful contact or refusal to participate in our survey; five substitutions were identified for each sampled physician. Substitution sampling followed the same stratification criteria used for the initial sample calculation. We controlled sample replacements for state, sex and age so that every physician who did not agree to participate was replaced by an individual with the same gender, age and specialty characteristics from the FCM list (see Equations (1) and (2)).

Equations (1) and (2): Sampling with Replacement Equations Employed.
(1)$$n=\frac{{z\alpha }^{2}*\left(p*q\right)}{d^{2}}=\frac{1.96^{2}*\left(0.5*0.5\right)}{0.05^{2}}$$
n = sample size;zα = critical value for the standard deviation (1.96);p = expected likelihood of the variable of interest within the population;q = adjustment for the expected prevalence of the variable of interest;d = sample error.
(2)$$n_{\mathrm{corr}}=\frac{N-n}{N-1}*n$$
n = sample size;N = physician population in the state.

Primary data were collected via a telephone survey carried out by data collectors, including a field coordinator, experienced interviewers and administrative staff responsible for checking missing data. Sample size calculations, sample selection, questionnaire design, substitution control, database assembly and data analysis were performed by the authors of this paper.

The questionnaire was previously field piloted and calibrated with interviewees to estimate the reposition rate. Reproducibility was tested by sampling a random sample after the field collection and repeating the interview, resulting in 100% agreement. The interviews consisted of a 30-min telephone questionnaire, containing 30 questions ranging from multiple, closed questions to interdependently concatenated and semi-open questions (see Appendix A).

### 2.2. Data Analysis

For this paper, we used survey variables related to physicians’ perceptions of changes in their workloads and earnings during the second year of the pandemic (see the Appendix A). Table 1 describes the variables used and their measurements.

In this study, the prevalences and 95% CIs of the variables of interest were estimated for each state in the sectors of practice studied (public, private and dual practice). Statistically significant differences at the 5% level were considered in the absence of overlapping 95% CIs. Median and interquartile range (Q1–Q3) are presented for the number of hours worked and for income in Brazilian reais. The database developed by the *Datafolha* data collectors was exported to R-Studio version 4.1.3 (R Foundation for Statistical Computing, Vienna, Austria) for statistical treatment.

### 2.3. Ethical Aspects

This study received approval from the Research Ethics Committees of the Federal University of Maranhão (CEP UFMA 3.051.875), and from the Faculty of Medicine of the University of São Paulo (CEP FMUSP 3.136.269), and was approved by Brazil’s FCM, which provided a list of physicians registered with the Councils of Medicine in the two states, and their respective telephone contacts.

All the physicians contacted were informed about the research objectives, and anonymity of the data and information provided was guaranteed by substituting names with codes. All the survey participants provided their informed consent (see the Appendix A).

## 3. Results

Among the 1183 physicians interviewed in the telephone survey, men predominated, and most of these were young men between 24 and 34 years of age. The respondents were equally distributed between rural and urban areas. The majority of respondents engage in dual practice; exclusive practice in the public sector being the next most common form of work reported by the respondents. Medical work takes place in different health establishments, but most of the respondents work in hospitals or clinics and perform surgical activities. Among those who reported changes in the number of hours worked, those who experienced an increase had a higher median (20; 10–30) than those who experienced a decrease (12; 8–20). On the other hand, among those who reported changes in income earned from their work, those who experienced a decrease had a higher median monthly salary (R$8000; 5000–10,000, or USD1487) than those who experienced an increase (6000; 5000–10,000, or USD1115) (Table 2).

When comparing the variables between the two states, there was a larger proportion of physicians in MA with the following characteristics: (a) age ranging from 35 to 44 years of age; (b) working in urban areas and in the capital; and (c) performing outpatient surgeries, without hospitalization or local anesthesia. On the other hand, in SP the prevalence of physicians working in the private sector was higher. The median increases in hours worked (24; 12–33) and income earned from work (9000; 5000–10,000, or USD1673) were higher in MA than in SP. (Table 2).

### 3.1. Impact of COVID-19 Pandemic on Labor Dynamics

At the time of the interview, more than two-thirds of physicians reported impacts on work dynamics that were still felt or that had become permanent (73.4%; 95% CI: 70.7–76.1). However, there were no statistically significant differences between the states, because the perceived impacts were equally great in MA and SP. Among physicians working in SP, the prevalence of these impacts was 74.9% (95% CI: 71.4–78.3), and in MA it was 71.5% (95% CI: 67.3–75.8) (Figure 1).

Physicians predominantly reported changes in the number of hours worked. Just over a third reported an increase in these hours, and another third reported a decrease. However, there were no differences between the states. In MA, there was a statistically significantly higher increase in hours worked in the public sector (44.4%; 95% CI: 38.0–50.8), followed by an equally high percentage of physicians who reported no change in their work dynamics (40.8%; 95% CI: 34.5–47.1). In SP, in the public sector, the proportion reporting that their hours remained the same was significantly higher (54.4%; 95% CI: 48.0–60.1), and those experiencing an increase in working hours had a lower prevalence (27.8%; 95% CI: 22.0–33.6) than that observed in MA. (Table 3).

Among physicians who work in the private sector, those who reported a reduction in the hours worked predominated in both states (SP: 48.3%; 95% CI: 41.7–54.9; MA: 48.9%; 95% CI: 42.3–55.5). Among those engaging in dual practice, the percentage reporting changes (increases or decreases) in the hours worked and the percentage reporting that their hours remained the same in both states were equivalent and showed no statistically significant difference (Table 3).

### 3.2. Changes in Physicians’ Incomes

There was a predominance of physicians who reported changes in the pattern of income earned from their work. Among these, the prevalence of those who reported a decrease was higher in both states (SP: 41.3%; 95% CI: 37.4–45.1; MA: 37.4%; 95% CI: 33.3–41.4). Another relevant proportion reported that their income remained the same, and this proportion was about the same in both states. The increase in the income pattern of physicians in the public sector was higher in MA (30.3%; 95% CI: 24.4–36.2) compared to SP (14.4%; 95% CI: 9.8–19.0). However, the majority of physicians in both states (MA: 49.3%; 95% CI: 42.9–55.7; and SP: 62.2; 95% CI: 56.0–68.4) reported that their income remained unchanged in the period.

Among physicians working in the private sector, there was a predominance of those who reported a reduced income pattern in both states (SP: 52.9%; 95% CI: 46.3–59.9; MA: 51.1%; 95% CI: 44.5–57.7). In both states, physicians engaged in dual practice showed equivalent percentage changes (increases and decreases) in income patterns, and equivalent percentages reported that income patterns remained the same, but there were no statistically significant differences between SP and MA (Table 4).

## 4. Discussion

In our study, similar proportions of doctors declared having maintained, increased or decreased their working hours during the pandemic. However, the workloads and earnings of those working exclusively in the public sector increased the most during the second year of the pandemic. The largest proportion of private-only doctors in the sample appeared to have seen a decrease in their working hours, whereas the largest proportion of public-only doctors in MA declared having seen an increase in their working hours. Earnings generally decreased for doctors working exclusively in the private sector for both states. A substantial proportion of dual practitioners declared having maintained their pre-pandemic levels of income, possibly by taking on additional public sector work.

Our findings are affected by a few limitations. This was a cross-sectional study of physicians’ perceptions of changes in hours and earnings during the second year of the pandemic, and their responses may have been influenced by recall bias [22]. We recognize cross-sectional surveys have limitations in their ability to explain the evolution of physicians’ workloads and earnings overtime; but cross-sectional surveys are also accepted in our discipline as important instruments to take a snapshot of ongoing, rapidly evolving circumstances [23]. As we interviewed a diverse cohort of professionals, it was not possible to identify a common, absolute measure of changes in physicians’ individual workloads and earnings, and perceptions of proportional increases and decreases were used instead. As it is not straightforward to distinguish between private and public providers in Brazil [12], our categories of public-only and private-only physicians were calculated based on the juridic nature of the employer, irrespective of whether the services were provided in public or private facilities. Therefore, our physician categories may not be directly comparable to those in similar studies elsewhere [24]. Finally, Maranhão and São Paulo represent very particular settings in terms of income distribution, organization of healthcare services and labor markets characteristics [13]; therefore, the study findings may not be entirely generalizable to the rest of Brazil, let alone to other low- and middle-income countries.

The decrease in working hours in the private sector, and the relative increase in the public sector, suggest that it is the public sector that has borne the largest burden of the pandemic in the two Brazilian states studied. This can be explained as follows: (a) COVID-19 affected a large share of the Brazilian population without private health plans [25], who had little choice but to seek care in public facilities; and (b) COVID-19 treatment often required hospitalization and intensive care services, which are often not available from private providers, which traditionally focus on outpatient care (such as walk-in clinics and small hospitals). This is consistent with the prevailing view that public health sectors play a key role in the fight against pandemics [26], and with what was observed in China [27] and India [28] during the pandemic—although in both countries nurses appear to have experienced the greatest impact in workload. In the pre-pandemic context, the private sector acted as an employer and provider of essential services. Dependence on public health services increased in the early stages of the pandemic, when many private facilities closed or severely curtailed their services [29]. This raises the issue of private sector involvement in the delivery of public goods, and how governments should harness private sector resources during health emergencies [30].

The decrease in private-sector-only doctors’ earnings in our sample is likely to have been a consequence of the decreased demand for—or suspension of—non-essential services during the pandemic. It is, however, significant that public sector earnings appear to have in fact increased in certain instances (particularly in MA), possibly as additional services were implemented because of the pandemic, and many newly graduated doctors were temporarily contracted to staff COVID-19 departments. In Brazil (as in other LMICs), public sector spending had been capped before the pandemic to reduce budget deficits and to contain inflationary pressures [31]; it will be crucial to monitor what happens to the new physicians and health services introduced to deal with COVID-19. Some commentators believe that health spending caps will be revised because of the pandemic [32], but the extent of such changes will most likely be influenced by the result of the upcoming presidential elections.

It is relevant that the dual practitioners in our sample managed to broadly maintain their pre-pandemic workloads and earnings, particularly as they represented the majority of doctors in our sample (61.6%), and represent the majority in the whole country [17]. It is most likely that they achieved this by compensating for the loss in hours and income from their private employment by taking on additional work in the public sector, where new opportunities were created during the pandemic. While this is consistent with economic theories regarding physician target income behavior [33], it is also possible that physicians may have taken advantage of telematic consultations to compensate for the loss of earnings, as was documented in China during the pandemic [34]. It will be important to explore the extent to which such services represent an additional form of dual practice in LMICs [35], and their consequences for physicians’ allocation of time across competing public and private engagements [36].

The effects of the pandemic on physicians’ workloads and earnings appear to have differed for MA and SP. For MA, our data show a general increase in working hours and earnings, mostly driven by public sector-only doctors. Although such a differential impact is consistent with what was witnessed in the two states during the pre-pandemic crisis [13], a recent study on risk factors suggests that the smaller number of physicians per capita in MA led to increased workloads, which increased physicians’ exposure to contracting the virus [37]. It will be important in the future to consider the fact that elements of health systems with fewer resources are affected unevenly by health emergencies, particularly as the increased workload (and associated working opportunities) is shared among a smaller pool of professionals.

Our findings have substantial implications for health policies in Brazil, and for the wider community of health policymakers in LMICs. As in our study, public-sector doctors appear to have been the enablers of the required surge in services and capacity during COVID-19, it is important for national governments and international institutions to invest in public health sectors worldwide as a means of developing epidemic preparedness [38]. In particular, it is crucial that the additional capacity and expenditures mobilized during COVID-19 are maintained, so as not to squander the progress made in strengthening fragile systems in resource-scarce settings. Although the role of the private sector in achieving universal health coverage objectives is increasingly being recognized [39], our findings confirm that new ways must be found to harness for-profit actors to deliver social benefits [40], as some private sector actors in Brazil do not seem to have contributed as much as public sector ones to the fight of the pandemic, despite recording significant profits [41]. New modalities for working remotely that have been scaled up during the pandemic are likely to support such processes [42].

It is not straightforward what the wider implications of our findings should be for health labor markets worldwide. On the one hand, the imbalances in demand and supply of physicians and salaries observed our study are likely to be overcome in the medium term. On the other hand, labor markets during external shocks, such as this pandemic, may need more flexible regulation to adjust more rapidly to the changing conditions. Either way, flexibility will always be a challenge in those countries with finite training capacity [43].

## 5. Conclusions

Although plentiful evidence has been accumulated on the impact of COVID-19 on the health of health workers, it is still unclear how their professional dynamics and livelihoods have been affected by the pandemic. We conducted a cross-sectional representative survey among the physician population in two states in Brazil, with the objective of understanding how their working hours and earnings changed during the second year of the pandemic.

The largest proportion of private-only doctors in our sample declared that they saw a decrease in their working hours, whereas working hours increased for the largest proportion of public-only doctors in MA. Earnings broadly remained stable in the public sector in the two states, although a substantial proportion of public-sector doctors in MA declared having seen an increase in their earnings, whereas these instead decreased for doctors working exclusively in the private sector for both states.

As public-sector doctors seem to have been the enablers of the required surge in capacity during COVID-19, we conclude that it is important for national governments and international institutions to invest in public health sectors in LMICs so as to develop epidemic preparedness, and to avoid squandering the progress made in strengthening fragile systems. New ways will also have to be found to harness for-profit actors to deliver social benefits, such as the provision of life-saving health services during emergencies.

## Figures and Tables

**Figure 1 ijerph-19-10085-f001:**
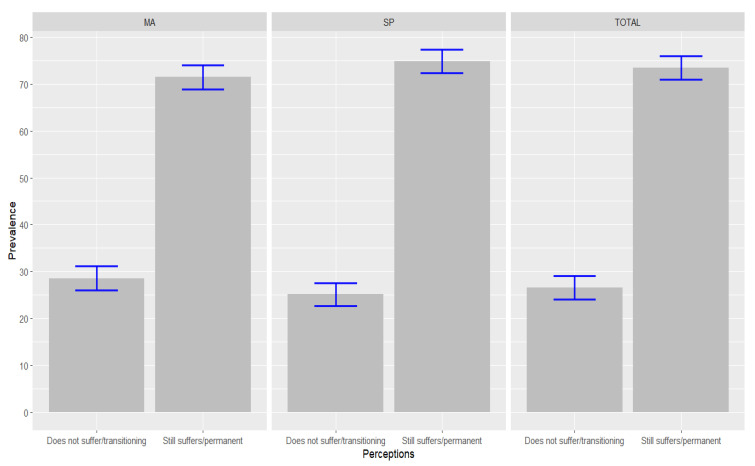
Perceptions of the impacts of the COVID-19 pandemic on work dynamics among physicians interviewed in SP and MA, Brazil, 2021. Blue lines: 95% Confidence Interval (95% CI). Source: Survey on the Labor Market and Impact of the New Coronavirus, in the States of SP and MA, Brazil, 2021.

**Table 1 ijerph-19-10085-t001:** Variables and their categories investigated among physicians interviewed in the Survey on the Labor Market and Impact of the New Coronavirus in SP and MA, Brazil, 2021.

Variables	Categories
Q.1 Considering your current medical work, with its routine, format, volume of patients, and working hours, when compared to a usual level of work, before March 2020:	It has not been impacted by the pandemic, performs the same work Has been impacted, but resumed what used to be done before Still suffers from impacts caused by the pandemic The pandemic brought changes that will be permanently incorporated in his/her work Physician’s work was interrupted due to the pandemic (is not currently working as a physician) OR Retired from/definitively abandoned physician work due to the pandemic
Q.3 (FOR EACH Q.2 = 1) In this medical activity _____ (READ THE ITEM), do you attend or did you attend, on a regular basis, only to patients with health insurance or private patients; attend/attended only Unified Health System (SUS) patients (either at a public, philanthropic facility, social organization or another type of facility that provides services to SUS) OR attend/attended to both types of patients, work in both public and private sectors?	Public Private Dual practice
Q.13 Considering a regular/habitual working week before the start of the pandemic, and your work after March 2020, has your number of worked hours per week increased, remained the same or decreased?	Increased Remained the same Decreased
Q.14a (IF Q13 = 1) By approximately how many weekly hours would you say your work has increased since the start of the pandemic? (SPONTANEOUS AND SINGLE ANSWER) Q.14b (IF Q13 = 3) By approximately how many weekly hours would you say your work has decreased since the start of the pandemic? (SPONTANEOUS AND SINGLE ANSWER)	_______ hours (WRITE DOWN). 9999. Does not know
Q.15 Considering your monthly salary before March 2020, at the start of the pandemic, has your salary increased, remained the same or decreased?	Increased Remained the same Decreased
Q.16a (IF Q.15 = 1) By how much, in reais, did your monthly salary increase?	_______ earnings (WRITE DOWN). 9999. Does not know
Q.16b (IF Q.15 = 3) By how much, in Reais, did your monthly salary decrease?	

Source: Survey on the Labor Market and Impact of the New Coronavirus in SP and MA, Brazil, 2021.

**Table 2 ijerph-19-10085-t002:** Socio-demographic characteristics of physicians interviewed in the Survey on the Labor Market and Impact of the New Coronavirus in SP and MA, Brazil, 2021.

Characteristics/Variables	Total (n = 1183)		SP (n = 632) 53.4% (50.6–56.3)		MA (n = 551) 46.6% (43.7–49.4)	
	%	95% CI	%	95% CI	%	95% CI
Gender						
Male	56.2	(53.4–59.0)	54.1	(50.2–58.0)	58.6	(54.5–62.7)
Female	43.8	(41.0–46.6)	45.9	(42.0–49.8)	41.4	(37.3–45.5)
Age						
24 to 34	34.1	(31.5–36.9)	34.3	(30.6–38.0)	33.9	(30.0–37.9)
35 to 44	24.5	(22.1–27.1)	20.7	(17.6–23.9)	28.9	(25.1–32.6)
45 to 59	20.4	(18.2–22.8)	22.3	(19.1–25.6)	18.1	(14.9–21.4)
≥60	21.0	(18.7–23.4)	22.6	(19.4–25.9)	19.1	(15.8–22.3)
Geographical location of deployment						
Rural areas (Interior)	50.5	(47.7–53.4)	54.9	(51.0–58.8)	45.6	(41.4–49.7)
Urban areas around capital cities	49.5	(46.6–52.3)	45.1	(41.2–49.0)	54.5	(50.3–58.6)
Health sector of deployment						
Exclusively private	12.9	(11.1–14.9)	14.3	(11.6–17.0)	8.2	(5.9–10.5)
Exclusively public	25.4	(23.0–28.0)	27.6	(24.1–31.1)	26.0	(22.3–29.8)
Dual practice	61.6	(58.8–64.4)	58.1	(54.4–62.0)	65.8	(61.8–69.8)
Employment in specific health services						
Outpatient clinical services (hospital or clinics)	82.7	(80.4–84.7)	81.2	(78.1–84.2)	84.2	(81.2–87.2)
Diagnostic tests, equipment-related services	33.1	(30.5–35.9)	32.0	(28.4–35.6)	34.5	(30.5–38.5)
Surgery (in-patient care)	38.8	(36.1–41.6)	36.9	(33.1–40.7)	41.0	(36.9–45.1)
Outpatient surgery	38.0	(35.3–40.8)	31.7	(28.1–35.3)	45.4	(41.2–49.6)
Administrative position	24.3	(22.0–26.9)	22.2	(19.0–25.4)	26.9	(23.2–30.6)
Teaching and research	26.4	(23.9–28.9)	28.2	(24.7–31.7)	24.3	(20.7–27.9)
Hours worked per week ^1^ (Median, Q1–Q3)						
Increased	20	(10–30)	18	(10–24)	24	(12–33)
Decreased	12	(8–20)	12	(10–20)	12	(8–20)
Earnings (in Brazilian Reais) ^2^ (Median, Q1–Q3)						
Increased	6000	(5000–10,000)	5000	(4000–10,000)	9000	(5000–10,000)
Decreased	8000	(5000–10,000)	8000	(5000–10,000)	8000	(5000–12,000)

Notes: ^1^ Only those who reported changes in hours worked were considered. ^2^ Only those who reported changes in labor income were considered. Q1: 25th percentile; Q3: 75th percentile.

**Table 3 ijerph-19-10085-t003:** Perceptions of the impacts of the COVID-19 pandemic on hours worked among physicians interviewed in SP and MA, Brazil, 2021.

Characteristics		Total (n = 1181)		SP (n = 632) 53.4% (50.6–56.3)		MA (n = 551) 46.6% (43.7–49.4)	
		%	CI 95%	%	CI 95%	%	CI 95%
Public	Hours worked						
	Increased	37.9	(31.7–44.1)	27.8	(22.0–33.6)	44.4	(38.0–50.8)
	Reduced	16.0	(11.3–20.7)	17.8	(12.9–22.7)	14.8	(10.2–19.4)
	Remained the same	46.1	(39.7–52.5)	54.4	(48.0–60.1)	40.8	(34.5–47.1)
Private	Increased	19.6	(14.3–24.9)	20.7	(15.3–26.1)	15.5	(10.7–20.3)
	Reduced	48.4	(41.8–55.0)	48.3	(41.7–54.9)	48.9	(42.3–55.5)
	Remained the same	32.0	(25.8–38.2)	31.0	(24.9–37.1)	35.6	(29.3–41.9)
Double practice	Increased	33.6	(30.2–37.0)	33.2	(29.8–36.6)	33.9	(30.5–37.3)
	Reduced	31.2	(27.8–34.6)	31.1	(27.7–34.5)	31.4	(28.0–34.8)
	Remained the same	35.2	(31.7–38.6)	35.7	(32.2–39.2)	34.7	(31.2–38.2)
Total	Increased	31.9	(29.2–34.5)	29.0	(25.4–32.5)	35.2	(31.2–39.2)
	Reduced	31.3	(28.7–34.0)	33.8	(30.2–37.6)	28.5	(24.7–32.2)
	Remained the same	36.8	(34.0–39.5)	37.2	(33.4–41.0)	36.3	(32.3–40.3)

**Table 4 ijerph-19-10085-t004:** Perceptions of the impacts of the COVID-19 pandemic on income patterns among physicians interviewed in SP and MA, Brazil, 2021.

Characteristics		Total (n = 1181)		SP (n = 632) 53.4% (50.6–56.3)		MA (n = 551) 46.6% (43.7–49.4)	
		%	95% CI	%	95% CI	%	95% CI
Public	Hours worked						
	Increased	24.1	(18.6–29.6)	14.4	(9.8–19.0)	30.3	(24.4–36.2)
	Reduced	21.6	(16.3–26.9)	23.3	(17.9–28.7)	20.4	(15.2–25.6)
	Remained the same	54.3	(47.9–60.7)	62.2	(56.0–68.4)	49.3	(42.9–55.7)
Private	Increased	12.3	(6.7–17.8)	12.1	(6.6–17.6)	13.3	(8.8–17.8)
	Reduced	52.5	(45.6–58.8)	52.9	(46.3–59.9)	51.1	(44.5–57.7)
	Remained the same	35.2	(28.9–41.5)	35.1	(28.8–41.4)	35.6	(29.3–41.9)
Double practice	Increased	19.7	(16.8–22.6)	19.4	(16.5–22.3)	20.0	(17.1–22.9)
	Reduced	41.5	(37.9–45.1)	40.3	(36.7–43.9)	42.8	(39.2–46.4)
	Remained the same	38.8	(35.3–42.3)	40.3	(36.7–43.9)	37.2	(33.7–40.7)
Total	Increased	19.2	(16.9–21.4)	16.8	(13.9–19.7)	22.0	(18.5–25.4)
	Reduced	39.5	(36.7–42.3)	41.3	(37.4–45.1)	37.4	(33.3–41.4)
	Remained the same	41.3	(38.5–44.1)	41.9	(38.1–45.8)	40.7	(36.5–44.8)

Source: Survey on the Labor Market and Impact of the New Coronavirus in SP and MA, Brazil, 2021.

## Data Availability

The data presented in this study are available on request from the corresponding author. The data are not publicly available due to privacy and anonymity guidelines.

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
