# Peer review of "The Impact of the COVID-19 Pandemic on Physicians’ Working Hours and Earnings in São Paulo and Maranhão States, Brazil"

_ijerph, 2022, doi:10.3390/ijerph191610085_

Round 1
Reviewer 1 Report
An interesting contribution to understand the characteristics of the medical labor market in Brazil. The text is easy to read and presents several useful elements for the health workforce research globally, including in the LMICs.
Main points of the study. What the authors' main ideas are and how they tie them together are not totally evident from either the abstract or the findings. Is it possible that doctors have not been significantly impacted by the pandemic, or have they been disproportionately affected in the public sector? Or possibly Maranhão doctors have had greater hardships in terms of workload and pay? I would recommend the authors to make the points made in the abstract more concise and clearer, and to prominently display them at the beginning of the discussion.
Background information on Brazilian doctors. There should be more information in the Introduction and Discussions regarding the unique characteristics of the medical labor market in Brazil and the part they played in the pandemic. It appears that the disparities between the health workforces in the states of Maranhão and São Paulo are not fully covered in the background section on pages 1 and 2, even though it is informative. The references listed below can help in illuminating these differences.
The study's pertinence to other LMICs. This is likely the area that requires the most development if the work is to be of interest to the readers of the journal. Although the unique characteristics of the Brazilian labor market are important for understanding the results, I think that further comparisons to other countries' experiences should be done to draw conclusions for the larger community of LMICs. Partially, the discussion already accomplishes this, but it might be improved by making similarities to what has been seen in China and India.
The theory's influence on the results. The theoretical implications of these Brazilian actual findings were not always obvious. The debate brought up the income behavior theory of physicians, but what are the key linkages to the theory for this study? Is the public sector—in this example, the SUS—required for pandemic readiness? Or perhaps it would be more accurate to state that the health labor markets aren't flexible enough? A few citations from the literature on human resources for health are provided below that may be used to explain these conceptual issues.
Andrade, Rodrigo de Oliveira. ‘Covid-19 Is Causing the Collapse of Brazil’s National Health Service’. BMJ 370 (30 July 2020): m3032. https://doi.org/10.1136/bmj.m3032.
Blumenthal, David, Elizabeth J. Fowler, Melinda Abrams, and Sara R. Collins. ‘Covid-19 — Implications for the Health Care System’. New England Journal of Medicine 383, no. 15 (8 October 2020): 1483–88. https://doi.org/10.1056/NEJMsb2021088.
Campbell, Jim, Gilles Dussault, James Buchan, Francisco Pozo-Martin, and Giorgio Cometto. ‘A Universal Truth: No Health Without a Workforce’. Forum Report, Third Global Forum on Human Resources for Health. Recife, Brazil: Global Health Workforce Alliance, 2013. ISBN 978 92 4 150676 2. https://www.who.int/workforcealliance/knowledge/resources/hrhreport2013/en/.
Costa, Danielle Conte Alves Riani, Ligia Bahia, Elza Maria Cristina Laurentino de Carvalho, Artur Monte Cardoso, and Paulo Marcos Senra Souza. ‘Oferta pública e privada de leitos e acesso aos cuidados à saúde na pandemia de Covid-19 no Brasil’. Saúde em Debate 44 (23 August 2021): 232–47. https://doi.org/10.1590/0103-11042020E415.
De Jaegher, Kris, and Marc Jegers. ‘A Model of Physician Behaviour with Demand Inducement’. Journal of Health Economics 19, no. 2 (1 March 2000): 231–58. https://doi.org/10.1016/S0167-6296(99)00029-6.
Gragnolati, Michele, Magnus Lindelow, and Bernard Couttolenc. ‘Twenty Years of Health System Reform in Brazil : An Assessment of the Sistema Único de Saúde’. Washington, DC: World Bank, 13 June 2013. https://doi.org/10.1596/978-0-8213-9843-2.
Haldane, Victoria, Chuan De Foo, Salma M. Abdalla, Anne-Sophie Jung, Melisa Tan, Shishi Wu, Alvin Chua, et al. ‘Health Systems Resilience in Managing the COVID-19 Pandemic: Lessons from 28 Countries’. Nature Medicine 27, no. 6 (June 2021): 964–80. https://doi.org/10.1038/s41591-021-01381-y.
Mills, Anne. ‘Health Care Systems in Low- and Middle-Income Countries’. The New England Journal of Medicine 370, no. 6 (6 February 2014): 552–57. https://doi.org/10.1056/NEJMra1110897.
Scheffler, Richard M. The Labour Market for Human Resources for Health in Low- and Middle-Income Countries. Geneva, n.d.
Author Response
Comment by reviewer: An interesting contribution to understand the characteristics of the medical labor market in Brazil. The text is easy to read and presents several useful elements for the health workforce research globally, including in the LMICs.
Comment by reviewer: Main points of the study. What the authors' main ideas are and how they tie them together are not totally evident from either the abstract or the findings. Is it possible that doctors have not been significantly impacted by the pandemic, or have they been disproportionately affected in the public sector? Or possibly Maranhão doctors have had greater hardships in terms of workload and pay? I would recommend the authors to make the points made in the abstract more concise and clearer, and to prominently display them at the beginning of the discussion.
Our response: Our key point is that some doctors have been affected more than others in the pandemic, notably for Brazil, those working in the public sector and in Maranhão. Private-sector-only doctors saw their workloads and earning decreased. Dual practitioners (the majority in our sample and in the country) managed to keep the same level of earnings, possibly by taking on additional public sector work. We have now made these messages more prominent in the abstract (Abstract, pag.1, Ln 20-28). (Discussion, pag.9, Ln 262-260).
Comment by reviewer: Background information on Brazilian doctors. There should be more information in the Introduction and Discussions regarding the unique characteristics of the medical labor market in Brazil and the part they played in the pandemic. It appears that the disparities between the health workforces in the states of Maranhão and São Paulo are not fully covered in the background section on pages 1 and 2, even though it is informative. The references listed below can help in illuminating these differences.
Our response: We have now used data from the latest Demografia Médica (2020) to describe the labour market differences between the two states, including data on salary levels in the two regions (Introduction pag. 2 and 3, Ln 94-105).
Comment by reviewer: The study's pertinence to other LMICs. This is likely the area that requires the most development if the work is to be of interest to the readers of the journal. Although the unique characteristics of the Brazilian labor market are important for understanding the results, I think that further comparisons to other countries' experiences should be done to draw conclusions for the larger community of LMICs. Partially, the discussion already accomplishes this, but it might be improved by making similarities to what has been seen in China and India.
Our response: We have now added comparisons and references with what observed in China and India in the Discussion (pag.9, Ln 292-296), and reflected on the overall implications for LMICs (pag.10, Ln 317-322).
Comment by reviewer: The theory's influence on the results. The theoretical implications of these Brazilian actual findings were not always obvious. The debate brought up the income behavior theory of physicians, but what are the key linkages to the theory for this study? Is the public sector—in this example, the SUS—required for pandemic readiness? Or perhaps it would be more accurate to state that the health labor markets aren't flexible enough? A few citations from the literature on human resources for health are provided below that may be used to explain these conceptual issues.
Our response: Thank you for these useful theoretical references. We have now elaborated in the Discussion on the role of the public sector and labour markets in epidemics on the basis of Scheffler et al (2016) work (pag. 10, Ln 353-359).

Reviewer 2 Report
Dear Authors
I have some questions and comments.
1. What does GDP mean?
2. The aim of the study doesn't correspond with results. It is necessarry to improve.
3. The section Discussion should include the comparison of health system organisation between Brasil and diffrent countries, it is necessary to compare physicians working hours in diffrent countries.
Author Response
Dear Authors
I have some questions and comments.
- Comment by reviewer: What does GDP mean?
Our response: ‘GDP’ stands for ‘Gross Domestic Product’. We have now spelled out the acronym early in the Introduction (pag.2, Ln 94)
- Comment by reviewer: The aim of the study doesn't correspond with results. It is necessarry to improve.
Our response: We have now expanded on the theoretical contribution of our empirical work from Brazil to labour economics (Discussion, pag. 10, Ln 353-358), so to make it more consistent with the paper’s stated objectives (Introduction, pag.3, Ln 118-120).
- Comment by reviewer: The section Discussionshould include the comparison of health system organisation between Brasil and diffrent countries, it is necessary to compare physicians working hours in diffrent countries.
Our response: We have now included examples, comparison and references to what observed in other LMICs, particularly in China and India in (Discussion, pag.9, Ln 296-300), and reflected on the overall implications for LMICs (pag.10, Ln 324-328).

Reviewer 3 Report
Remarks:
1) Why did the authors only select two states? In the text, they explain that. "With different socioeconomic realities and different levels of access to the health system" Can the research results be representative so that the research results cover the whole of Brazil and other countries (low- and middle-income countries)? Why wasn't the other Brazilian state included in the study?
2) The research sample concerned approximately 0.4% of SP physicians and 7.2% of MA physicians - why was such a small percentage examined in SP? How were doctors of different medical specialties divided?
3) Although the authors wrote: (see Survey Questionnaire in the Annex 1) - there is no annex to the manuscript. (lines 129, 152)
4) Physicians should be examined at different times as the increase in COVID incidence has varied from week to month.
5) There are serious doubts about the research - its scope, questions, methodology, test sample There is no comprehensive description of the health service in Brazil, number of hospitalizations, COVID patient statistics etc.
6) Weakness in the use of statistical tools and the search for relationships between e.g. the number of patients, earnings, public spending and the effectiveness of the health care system in Brazil.
I do not recommend the text for further work.
Author Response
Remarks:
1) Comment by reviewer: Why did the authors only select two states? In the text, they explain that. "With different socioeconomic realities and different levels of access to the health system" Can the research results be representative so that the research results cover the whole of Brazil and other countries (low- and middle-income countries)? Why wasn't the other Brazilian state included in the study?
Our response: We explained upfront that this survey is part of a wider and specific UKRI study on the impact of the economic crisis and COVID-19 on São Paulo and Maranhão, two very different states in Brazil (Material and methods, pag 3, Ln 123-125). We recognize in the Discussion that the findings are valid for the two case-studies (internal validity), but that the external validity is limited (Discussion pag.10, Ln 284-288).
2) Comment by reviewer: The research sample concerned approximately 0.4% of SP physicians and 7.2% of MA physicians - why was such a small percentage examined in SP? How were doctors of different medical specialties divided?
Our response: This is an important point. The reason why a smaller proportion of physicians in São Paulo was selected is because the universes of physicians in the two states are very different in size: in SP we have 30% of all the physicians in Brazil, and in MA a little more than 1%. This is why we should not have a proportional sample from each state, as the MA one would be too small to allow for sufficient statistical N for some of the strata and doctors’ characteristics we are interested in. In our long experience in running surveys in Brazil, this is how we have resolved the issue of representativeness when comparing very different universes of physicians: we usually calculate two independent samples (one for each state), each reflecting the characteristics of interest for our stratification. Then, comparisons between the physicians in the two states are adjusted by weight, or by using a mixed model, to account for the different variance in the two sets. We have now provided this explanation for our sampling strategy in the Methods section (pag.3, Ln 133-146).
3) Comment by reviewer: Although the authors wrote: (see Survey Questionnaire in the Annex 1) - there is no annex to the manuscript. (lines 129, 152)
Our response: Apologies for this oversight. The complete survey questionnaire has now been uploaded in the submission system as Annex 1.
4) Comment by reviewer: Physicians should be examined at different times as the increase in COVID incidence has varied from week to month.
Our response: Please note this manuscript is based on a cross-sectional survey conducted during the second year of the pandemic; such design may have weaknesses, but cross-sectional surveys are commonly accepted in our discipline as important instruments to take a snapshot of ongoing, rapidly evolving circumstances. We have now acknowledged this limitation in the opening paragraph of the Discussion (pag.9, Ln 271-277).
5) T Comment by reviewer: there are serious doubts about the research - its scope, questions, methodology, test sample There is no comprehensive description of the health service in Brazil, number of hospitalizations, COVID patient statistics etc.
Our response: We have now tried to improve the clarity of the study’s scope, questions, methodology and sampling (see the responses to the points above). We have also expanded and developed the background section on Brazil’s health system, and the epidemiological impact of COVID-19 (pag.2, Ln 59-88).
6) Comment by reviewer: Weakness in the use of statistical tools and the search for relationships between e.g. the number of patients, earnings, public spending and the effectiveness of the health care system in Brazil. I do not recommend the text for further work.
Our response: We are sorry this reviewer did not value the first version of our work. We have now tried to improve the methodological section, the objectives of the paper, its limitations, and clarified the points above, as suggested also by the more positive comments of the other reviewers.

Round 2
Reviewer 2 Report
I don't have.
Reviewer 3 Report
The authors tried to prepare corrections to the text, which I assess positively. Unfortunately, I believe that authors need to rethink the entire concept of the manuscript, including the research methodology, research objectives, and research tools used.
I don't recommend for publication.